# Metatranscriptome Profiling of a Specialized Microbial Consortium during the Degradation of Nixtamalized Maize Pericarp

José Germán Serrano-Gamboa,[a] Carlos Abraham Fócil-Espinosa,[a] Pedro J. Cabello-Yeves,[b] Felipe H. Coutinho,[b] Rafael Antonio Rojas-Herrera,[a] Mónica Noel Sánchez-González[a]

aFacultad de Ingeniería Química, Universidad Autónoma de Yucatán, Mérida, México
bEvolutionary Genomics Group, Departamento de Producción Vegetal y Microbiología, Universidad Miguel Hernández, Alicante, Spain

**ABSTRACT** Lignocellulose degradation by microbial consortia is multifactorial; hence, it must be analyzed from a holistic perspective. In this study, the temporal transcriptional activity of consortium PM-06, a nixtamalized maize pericarp (NMP) degrader, was determined and related to structural and physicochemical data to give insights into the mechanism used to degrade this substrate. Transcripts were described in terms of metabolic profile, carbohydrate-active enzyme (CAZyme) annotation, and taxonomic affiliation. The PM-06 gene expression pattern was closely related to the differential rates of degradation. The environmental and physiological conditions preceding high-degradation periods were crucial for CAZyme expression. The onset of degradation preceded the period with the highest degradation rate in the whole process, and in this time, several CAZymes were upregulated. Functional analysis of expressed CAZymes indicated that PM-06 overcomes NMP recalcitrance through modular enzymes operating at the proximity of the insoluble substrate. Increments in the diversity of expressed modular CAZymes occurred in the last stages of degradation where the substrate is more recalcitrant and environmental conditions are stressing. Taxonomic affiliation of CAZyme transcripts indicated that *Paenibacillus macerans* was fundamental for degradation. This microorganism established synergistic relationships with *Bacillus thuringiensis* for the degradation of cellulose and hemicellulose and with *Microbacterium*, *Leifsonia*, and *Nocardia* for the saccharification of oligosaccharides.

**IMPORTANCE** Nixtamalized maize pericarp is an abundant residue of the tortilla industry. Consortium PM-06 efficiently degraded this substrate in 192 h. In this work, the temporal transcriptional profile of PM-06 was determined. Findings indicated that differential degradation rates are important sample selection criteria since they were closely related to the expression of carbohydrate-active enzymes (CAZymes). The initial times of degradation were crucial for the consumption of nixtamalized pericarp. A transcriptional profile at the onset of degradation is reported for the first time. Diverse CAZyme genes were rapidly transcribed after inoculation to produce different enzymes that participated in the stage with the highest degradation rate in the whole process. This study provides information about the regulation of gene expression and mechanisms used by PM-06 to overcome recalcitrance. These findings are useful in the design of processes and enzyme cocktails for the degradation of this abundant substrate.

**KEYWORDS** lignocellulose degradation, microbial consortium, metatranscriptome, nixtamalized maize pericarp, carbohydrate-active enzyme expression

L ignocellulosic residues derived from agroindustry are abundant and low-cost resources with the potential to produce second-generation biofuels and other value-added products (1, 2). Tortillas, a staple food in the Mexican diet, are produced

Address correspondence to Mónica Noel Sánchez-González, monica.sanchez@correo.uady.mx.

The authors declare no conflict of interest.

in a process called nixtamalization, where maize grains are cooked under alkaline conditions. The tortilla industry annually generates approximately 14,400,000 $m^3$ of wastewater containing about 300,000 tons of nixtamalized maize pericarp (NMP) (3). Although some pericarp components are lost during nixtamalization, NMP is an important source of fermentable sugars and fine chemicals (4).

Transformation of agro-industrial residues is limited by the recalcitrance of lignocellulose structures. In nature, microbial communities are involved in the decomposition of lignocellulosic substrates as part of the carbon cycle. Through the enrichment of environmental samples, it is possible to obtain specialized microbial consortia with desired degradative properties (5, 6).

The use of specialized lignocellulose-degrading consortia is a promising approach because selection enhances the carbohydrate-active enzyme (CAZyme) repertoire, achieving efficient decomposition of recalcitrant substrates (7, 8). During degradation, notable changes occur under environmental conditions, such as pH, cell and protein concentration, nutrient availability, and substrate composition among others (9–11).

Temporal metagenomic and metatranscriptomic analyses contribute to the understanding of microbial interactions, enzyme mechanisms, and functions under particular environmental conditions along a degradation process (7, 12–15). To obtain representative samples, the selection criteria are an important task of transcriptomic analysis. In temporal studies, the selection of samples is frequently based on degradation stages (12, 13) or enzyme activities (14). The stages of degradation are usually defined based on substrate consumption rates and calculated using the inoculation time as a reference (integral rates) (7, 12). However, if rates are calculated using the preceding time as a reference (differential rates), even small changes in the speed of substrate consumption can be determined.

Consortium PM-06, a specialized community enriched from NMP endogenous microbiota, degrades 87% of this tortilla residue in 192 h. *Aneurinibacillus migulanus*, *Paenibacillus macerans*, *Bacillus coagulans*, *Microbacterium* sp. LCT-H2, and *Bacillus thuringiensis* are the most abundant species in this community, comprising 86% of the population (16). Although PM-06 is an efficient NMP degrader, there is scarce information about the role of CAZymes in the process.

In this study, a temporal analysis of the transcriptional activity of consortium PM-06 during the degradation of NMP was performed. Samples were selected based on differential degradation rates under the hypothesis that the expression of CAZymes is upregulated in times preceding a high degradation rate period. Transcripts of selected samples were described in terms of metabolic profile, CAZyme annotation, and taxonomic affiliation. Moreover, the transcriptional CAZyme profile was related to structural and physicochemical data to give insights into the mechanism used for the degradation of NMP.

## RESULTS

**Microbial growth and degradation kinetics.** Consortium PM-06 degraded 81% of NMP solids after 192 h (Fig. 1a). Differential degradation rates varied along the process, obtaining the highest values during the first 8 h (~20 mg/h). Two additional increments were determined at 24 h (7.7 mg/h) and at 168 h (4.6 mg/h) (Fig. 1a). At the beginning of the process, NMP contained 15.3% cellulose, 23.7% xylan, 12.6% arabinose substituents, 33% starch, 4.4% lignin, and 11% of other components (ashes, extractives, uronic acids, and protein). Hemicellulose (arabinan and xylan) was consumed faster than cellulose and lignin. The fastest cellulose consumption was determined from 144 to 168 h, where concentration decreased 26.7% (Fig. 1b). At the end of the process, NMP residual solids contained more lignin than the initial substrate. Cell concentration exponentially increased after 4 h of incubation, reaching a maximum (0.31 mg/mL) at 72 h (Fig. 1c). After this time, the number of cells decayed. Selection of samples for metatranscriptomic analysis was based on differential degradation rates. Therefore,

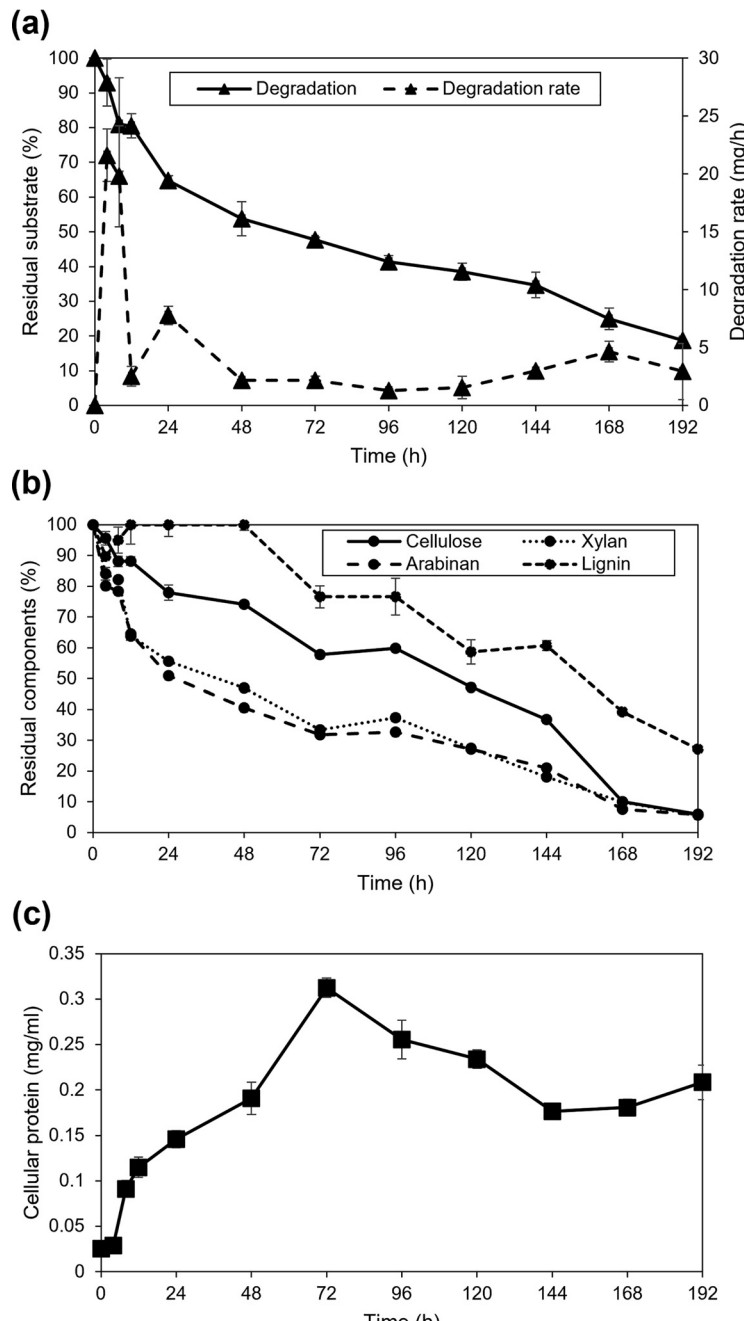

**FIG 1** NMP degradation and microbial growth kinetics. (a) NMP solids consumption and differential degradation rates. (b) Degradation profile of NMP components. (c) PM-06 microbial growth in terms of cellular protein concentration.

samples from high differential degradation rate periods (4 and 24 h) and from times preceding a high-rate period (0 h and 120 h) were selected.

**Reference metatranscriptome overview.** A reference metatranscriptome containing a total of 14,257 contigs was obtained by *de novo* cross-assembly of 308 million high-quality reads from all samples. The reference metatranscriptome contained 48,516 transcripts encoding 44,758 predicted proteins. Taxonomic annotation showed that transcripts were assigned mainly to the Bacteria domain (97.34%). *Firmicutes* and *Actinobacteria* were the dominant phyla (50.9 and 45. 8%, respectively), and at the species level (sequence identity of ≥99%), the predicted proteins were largely associated

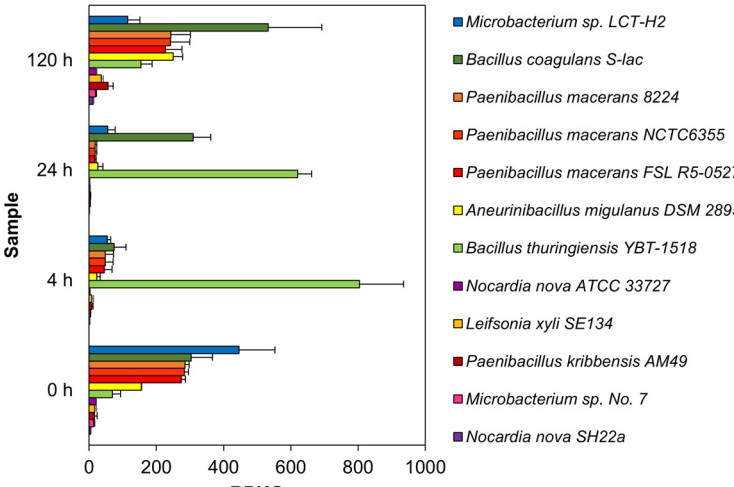

**FIG 2** Transcriptional activity of consortium members estimated by recruitment analysis of metatranscriptomic reads against 12 bacterial genomes. Genome coverage in each sample is expressed in reads per kilobase of genome per gigabase of metatranscriptome (RPKG).

with *Aneurinibacillus migulanus* (13.26%), *Paenibacillus macerans* (12.89%), *Nocardia nova* (9.98%), *Bacillus coagulans* (6.90%), *Microbacterium* sp. LCT-H2 (4.63%), *Bacillus thuringiensis* (2.42%), and *Leifsonia xyli* (2.09%). All identified taxa were consistent with those previously identified in the PM-06 metagenome (16).

**Transcriptional activity of consortium members.** The transcriptional activity of microorganisms was determined through a recruitment analysis (Fig. 2). For this purpose, the parameter reads per kilobase of genome per megabase of metatranscriptome (RPKG) was estimated by calculating the coverage of the mRNA reads against genomes of the most abundant species and according to taxonomic annotation of the reference metatranscriptomic and metagenomic data.

*Microbacterium, Paenibacillus, Bacillus,* and *Aneurinibacillus* were the most transcriptionally active genera with RPKG values greater than 50. At 0 h, the genome of *Microbacterium* sp. LCT-H2 recruited a larger set of mRNAs (445.62 RPKG), followed by *Bacillus coagulans* S-lac (303.07 RPKG) and *Paenibacillus macerans* 8224 (285.09 RPKG). In contrast, the transcriptional activity at 4 h was predominantly attributed to *Bacillus thuringiensis* YBT 518 (803.61 RPKG). At 24 h, genomes of *Bacillus thuringiensis* YBT 518 and *Bacillus coagulans* S-lac recruited most of the metatranscriptome reads (619.98 RPKG and 309.34 RPKG, respectively). Finally, at 120 h *Bacillus, Paenibacillus,* and *Aneurinibacillus* were the most transcriptionally active genera.

**Metabolic profile.** Functional analysis using KEGG orthology annotation identified the most represented pathways (Fig. 3). Metabolism of carbohydrates, genetic information processing, and environmental information processing were the most representative functions in all samples. The distribution of KEGG categories changed with degradation, indicating metabolic shifts. At 4 h, the proportion of transcripts associated with carbohydrate metabolism, genetic information processing, and energy metabolism increased and reached the highest values (16.29%, 21.75%, and 6.37%, respectively) in the whole process, indicating a stage of large uptake of carbon sources and preparation for exponential growth. In contrast, functions associated with metabolism of cofactors and vitamins decreased by one-third compared to 0 h. Lipid, nucleotide, and amino acid metabolism showed similar proportions in all samples.

**Global pattern of CAZyme genes.** A total of 1,221 coding sequences (2.7% of predicted proteins) were identified as putative CAZymes; however, only 555 were consistently annotated by the three tools provided by the automated CAZy annotation database (dbCAN; Data S1A in the supplemental material). From these sets, glycosyl hydrolase (GH) was the most abundant group (56.58%), followed by glycosyl transferases

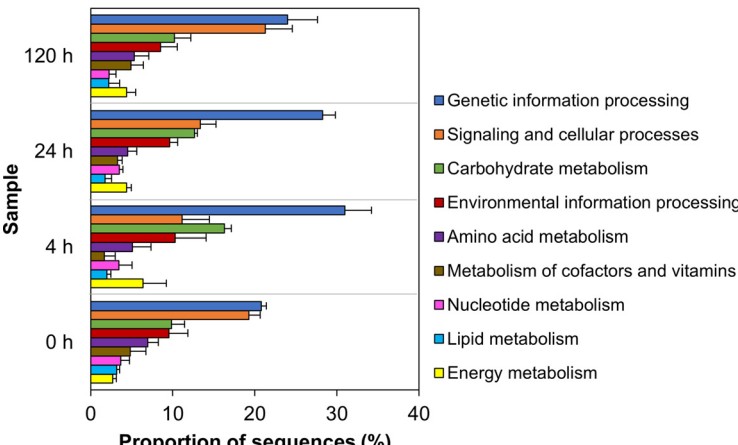

**FIG 3** Metabolic profile of transcripts according to KEGG orthology functional categories. Percentages correspond to the mean of retrieved mapped sequences in each sample (*n* = 3). The average proportion of the categories shown was greater than 1%. Error bars represent the standard deviation.

(GT) (27.93%) and carbohydrate esterases (CE) (11.17%). Coding sequences for carbohydrate-binding modules (CBM), polysaccharide lyases (PL), and auxiliary activities (AA) were identified in smaller proportions (Data S1A). The taxonomic affiliation of CAZymes was dominated by *Firmicutes* (62%) and *Actinobacteria* (33%) phyla. At the genus level, *Paenibacillus*, *Bacillus*, *Microbacterium*, and *Leifsonia* were the major contributors to the expression of different CAZyme families.

**Modular CAZymes in NMP degradation.** PM-06 exhibited transcripts encoding CAZymes with multidomain architecture involved in the decomposition of the structural components of NMP (Table S1). Identified catalytic domains associated with CBMs included several hemicellulases (GH10, GH26, GH43, GH44, and GH81), cellulases (GH5, GH6, GH9, GH48, and GH74), and a lytic polysaccharide monooxygenase (LPMO) belonging to the AA10 family. Except for AA10, which is taxonomically affiliated with *Bacillus cereus*, all modular CAZymes were expressed by *Paenibacillus macerans* (Table 1).

CBM3 was the most frequent among all expressed CBMs. The catalytic domains associated with this module included GH5 and GH6 (endoglucanases), GH48 (exoglucanase), and GH10 (endoxylanase). CBM59 (mannan/xylan/cellulose-binding) was associated with GH10, suggesting a versatile role of this enzyme. The architecture of LPMO included CBM5 with affinity to crystalline cellulose and complex polysaccharides (17, 18).

CBM6, CBM13, and CBM66 (involved in the recognition of substituted xylan [19]) were identified in two different multimodular enzymes containing GH43 (xylosidase/arabinofuranosidase) as the catalytic domain. CBM32 and CBM35 participated in different domain architectures targeting hemicellulose in association with GH26, GH44, or GH81.

**Taxonomic affiliation of differentially expressed transcripts.** The differential expression analysis performed to determine variations in transcriptional activity identified 21,000 differentially expressed (DE) transcripts (Fig. S2). A total of 407 genes encoding putative CAZymes were DE. From this set, all genes annotated as GT and CBM (unassociated) and/or affiliated with low-abundant microbial genera were manually removed. Finally, 181 CAZymes were selected for further analysis (Data S1B in the supplemental material).

DE transcripts were taxonomically affiliated and grouped according to preference over oligosaccharides, cellulose, hemicellulose, other carbohydrates, acetylated carbohydrates, and crystalline cellulose (Fig. 4) (DE transcript groups are defined in Data S1B). *Paenibacillus* and *Microbacterium* transcripts presented the highest expression levels ($\sim$10$^5$ counts per million [CPM]) at 0 h. *Paenibacillus* expressed a diverse set of CAZymes, including cellulases, hemicellulases, and oligosaccharidases. *Microbacterium*, *Leifsonia*, and *Nocardia* expressed a smaller and less diverse set of CAZymes, which was dominated by oligosaccharidases and other glycanases. At 4 and 24 h, *Bacillus* showed

**TABLE 1** Predicted structure and function of modular CAZymes identified in the PM-06 metatranscriptome

| Taxonomic affiliation and time of major expression | CAZyme annotation and architecture[a] | Functional characterization (reference) |
|---|---|---|
| *Bacillus cereus* 4 h | CBM5 AA10<br>4　21 41　208 455 | Lytic polysaccharide monooxygenase (59) |
| *Paenibacillus macerans* 120 h | GH5_1 CBM3<br>46　360 489　556 626 | Endoglucanase (60) |
| *Paenibacillus macerans* 0 h, 120 h | GH5_4 CBM46<br>68　351 379　465 575 | $\beta$-1,3-1,4-Endoglucanase (61) |
| *Paenibacillus macerans* 0 h, 120 h | GH6 CBM3<br>61　407 751　817 889 | Cellobiohydrolase (60) |
| *Paenibacillus macerans* 0 h | GH10 CBM3<br>82　363 534　613 686 | Endo-1,4-$\beta$-xylanase (62) |
| *Paenibacillus macerans* 120 h | GH10 CBM59<br>47　314 339　482 484 | Endo-1,4-$\beta$-xylanase (63) |
| *Paenibacillus macerans* 120 h | GH48 CBM3<br>64　522 872　952 1016 | Cellobiohydrolase (64) |
| *Paenibacillus macerans* 0 h | GH74 CBM3<br>99　717 903　982 1045 | Xyloglucanase (60, 64) |
| *Paenibacillus macerans* 120 h | GH9 CBM3 CBM3<br>40　474 501　586 766　847 911 | Endoglucanase (65) |
| *Paenibacillus macerans* 0 h, 120 h | CBM35 GH26 CBM3<br>44　162 178　512 720　800 863 | Mannanase (60) |
| *Paenibacillus macerans* 120 h | GH43 CBM6 CBM13 CBM13<br>41　355 378　506 508　587 610　568 | $\alpha$-L-Arabinofuranosidase/ $\beta$-xylosidase (19) |
| *Paenibacillus macerans* 0 h | GH43 CBM66 CBM6 CBM66<br>28　294 323　479 508　642 657　822 | $\alpha$-L-Arabinofuranosidase (64) |
| *Paenibacillus macerans* 120 h | GH44 CBM35 GH26 CBM3<br>43　552 660　775 792　1092 1208　1289 | Endoglucanase/xylanase (60) |
| *Paenibacillus macerans* 0 h | CBM32 CBM32 GH81 CBM32<br>57　171 216　323 365　1023 1119　1233 | $\beta$-1,3-Glucanase (66) |

[a]Enzyme architectures are schematically (not to scale) represented. Numbers indicate amino acid positions of predicted domains.

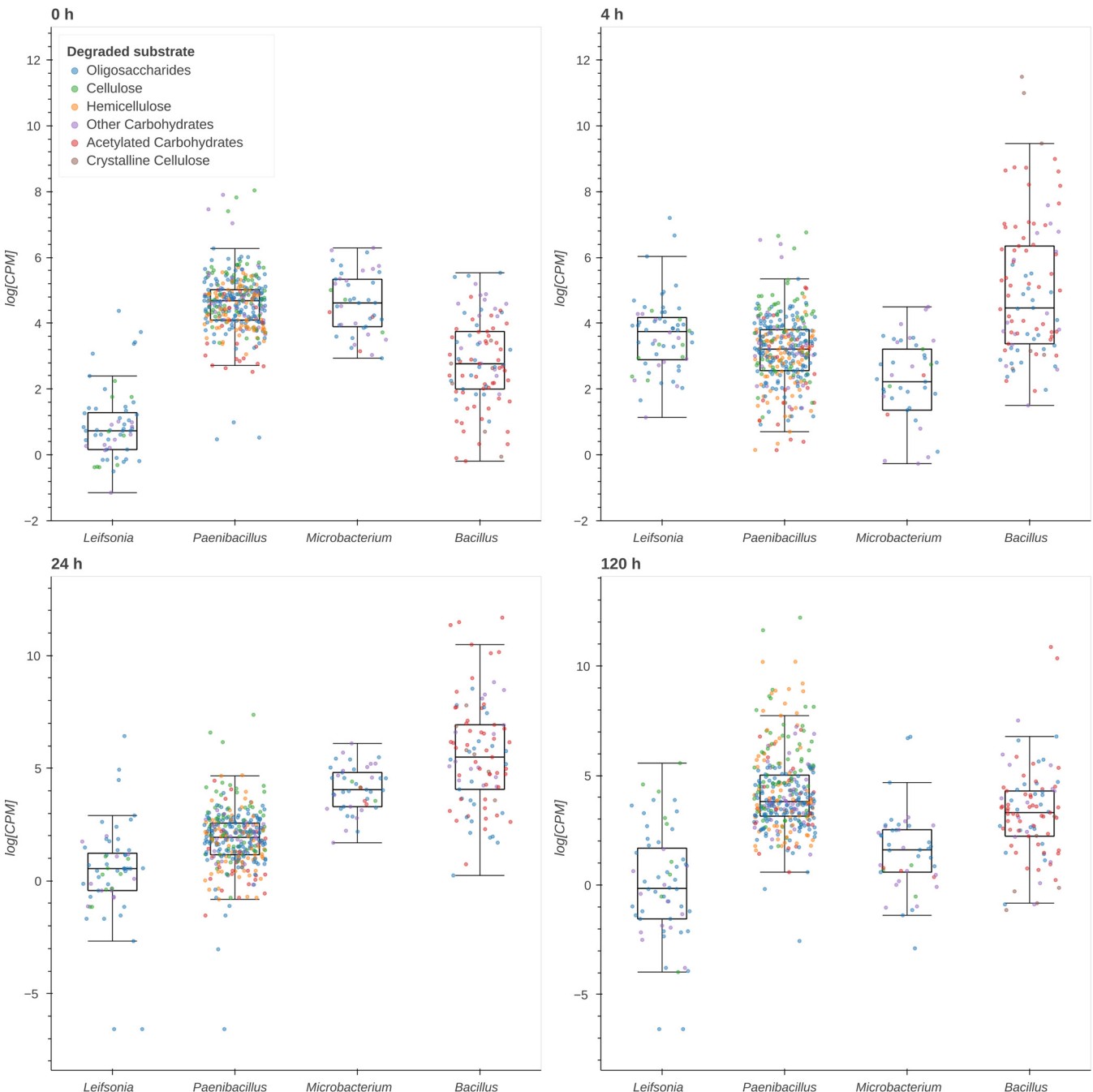

**FIG 4** Taxonomic affiliation of differentially expressed transcripts grouped according to substrate preference. Expression values were estimated as log CPM.

high levels of expression ($\sim 10^8$ CPM) of a carbohydrate esterase associated with hemi-cellulose deacetylation, an LPMO, and other CAZymes not directly related to the degradation of lignocellulose (i.e., amylases). Although *Paenibacillus* expression levels were low in these times ($\sim 10^3$ CPM), they included a broad set of enzyme functionalities. At 120 h, *Paenibacillus* CAZymes targeting cellulose and hemicellulose reached $10^{10}$ CPM, the maximum expression value of the whole process. The expression levels of *Microbacterium* and *Leifsonia* were lower than those determined at 0 h.

**Differentially expressed CAZymes involved in NMP degradation.** A heatmap was generated to visualize gene expression pattern and clustering (Fig. 5a). The selection of CAZyme transcripts was based on the activity over NMP components, expression values (transcript abundance above $1 \times 10^4$ CPM), and functional domains. The

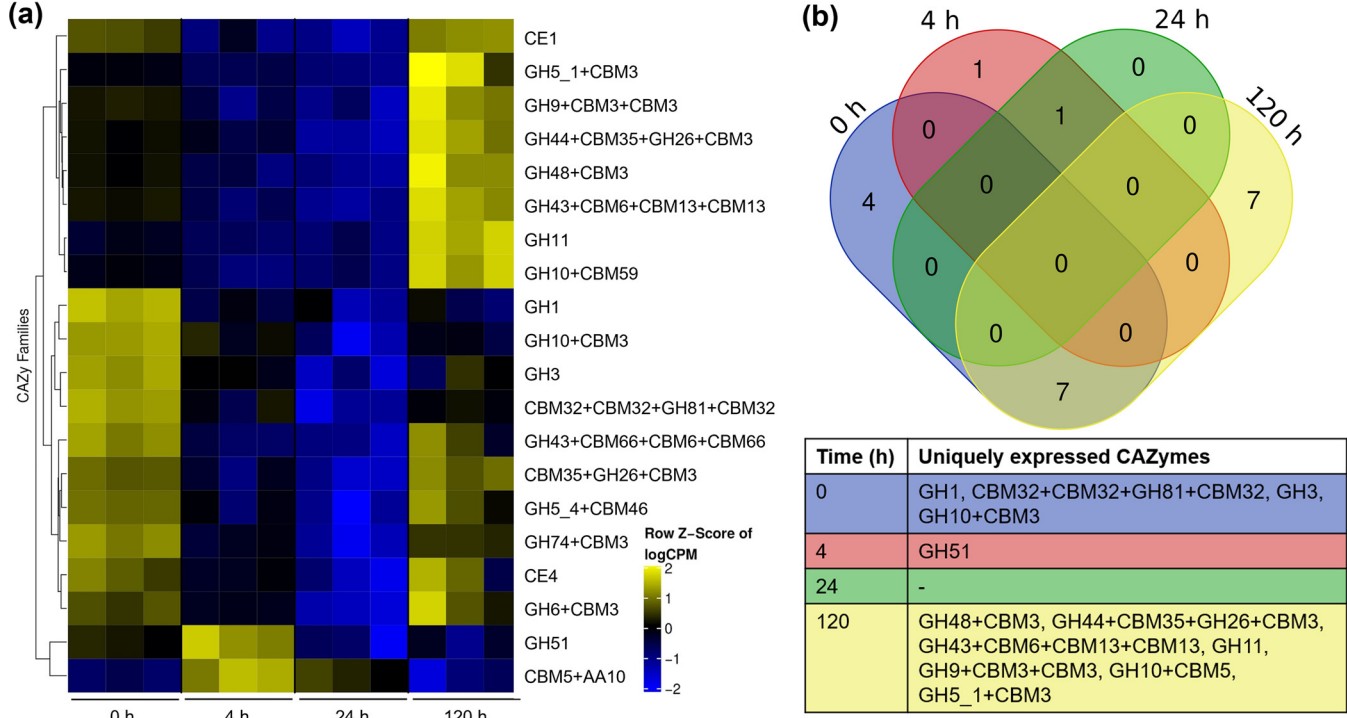

**FIG 5** Differential expression analysis. (a) Heat map of differentially expressed CAZyme genes (row *Z*-score of normalized CPM values) involved in NMP degradation. (b) Venn diagram of upregulated CAZymes (average *Z*-score of ≥0.4), indicating shared and unique families between time samples.

hierarchical clustering of families was performed based on the *Z*-score computed across samples. Clustering revealed two groups with similar expression profiles. The first contained 0- and 120-h samples, and the second included 4- and 24-h transcripts. This result correlated with the pattern observed in the recruitment analysis (Fig. 2).

A total of 11 and 14 CAZymes were overexpressed at 0 and 120 h with only 7 coincidences. Conversely, at 4 and 24 h, only two and one CAZyme, respectively, were upregulated. (Fig. 5b). CAZymes with CE1, CE4, GH3, GH10, GH26, and GH43 domains, related to hemicellulose decomposition (20, 21) and saccharification of solubilized oligosaccharides (22, 23), were upregulated at the onset of the degradation process. Likewise, significant expression of GH5 (average of $5.8 \times 10^4$ CPM) and GH6 ($4.2 \times 10^4$ CPM) families, associated with cellulose hydrolysis, was observed. (Fig. 5a).

At 4 h, the transcription of CAZymes decreased, and only GH51 (arabinofuranosidase) and AA10 (LPMO) were upregulated. All CAZyme families were downregulated at 24 h. The upregulated set of CAZyme genes at 120 h included GH5, GH9, GH10, and GH11 genes, which hydrolyze internal bonds of cellulose and xylan (24, 25). A notable expression of GH48 ($1.32 \times 10^5$ CPM), an exoglucanase with a prominent role in the decomposition of cellulose, was determined at this time (26). The high expression of CE1 ($1.6 \times 10^5$ CPM) and CE4 ($7.4 \times 10^4$ CPM) genes indicated the deacetylation of hemicellulose.

## DISCUSSION

In nature, lignocellulose degradation is a complex and dynamic process that involves the participation of different enzymes working in constant interaction. The amount and diversity of the enzymes produced depend on the source of the consortium, enrichment and selection conditions, and substrate structure and composition. In addition to CAZymes, the transformation of complex polysaccharides involves biochemical and physiological responses, many of them still unknown. In this study, the temporal expression of CAZyme genes by consortium PM-06 was analyzed and associated with environmental conditions and substrate composition changes to get insight into the mechanism used for the degradation of NMP.

PM-06 degraded NMP with different rates, obtaining the highest rate during the initial 8 h. The results indicated a relationship between the differential degradation rates and the transcriptional activity. In times preceding high degradation rate periods (0 and 120 h), the transcriptional activity was divided between different microbial species. The participation of distinct microorganisms suggests a synergistic behavior between degraders, cooperating for the production of enzymes with complementary functions in the so-called niche partition (27). Although at 0 and 120 h, several CAZyme genes were upregulated, there were few coincidences, suggesting differences in the regulation process.

Interestingly, at 0 h, the lowest concentration of cells was determined; however, a high number of CAZymes was overexpressed. *Paenibacillus* and *Microbacterium* were the main contributors to the expression of CAZymes related to the saccharification of polysaccharides and oligosaccharides, respectively. The initial stages of microbial growth have been associated with dramatic changes in the proteome and transcriptome for the production of digestive enzymes, biomass building, and preparation for cell division (28). In some lignocellulose degradation processes, the initial times have been recognized as important for degradation (14); however, there is scarce information about the CAZymes expressed, mainly because there are technical difficulties associated with the small number of cells (29). Consortia stabilization includes sequential transfers that select the population and accelerate the adaptation of bacteria to new environments (30). In this rapid adaptation, bacteria use strategies such as the "poised state" of the RNA polymerase, participation of the transcription initiation factor $\sigma$70, or the two-component system to transduce information from the environment between other mechanisms (28, 31, 32). The complex control regulatory networks responsible for predictable and sustainable responses to predefined inputs are conceptually related to biological memory (33, 34). Biological memory can be defined as "enduring changes in the mechanism of behavior based on prior experiences with environmental inputs" (35). Therefore, the high transcriptional levels at 0 h could be the result of biological memory restoration mechanisms. More studies are necessary to determine the regulation mechanisms involved in CAZyme transcription at the onset of degradation.

The population density at 120 h was higher than at 0 h; however, microorganisms were in the decay stage, probably because of a low concentration of soluble sugars. The high degradation rates obtained during the initial 48 h generated soluble sugars that supported exponential growth of the community (Fig. 1c). After this period, degradation rates decreased, probably because of the presence of a more recalcitrant substrate (increased amount of lignin), causing starvation. These conditions may induce the expression of a high number of CAZymes. In this time, like at 0 h, CAZymes were expressed mainly by *Paenibacillus* and *Microbacterium*.

At 4 and 24 h, degradation rates increased, and the expression of lignocellulose-targeted CAZymes was downregulated. The degradative activity incremented the concentration of soluble sugars, reducing the expression of CAZymes probably by carbon catabolite repression (36). Moreover, the presence of easily assimilable sugars could decrease the metabolic cooperation between consortia members (27). Therefore, in these times, the transcriptional activity was mainly devoted to *Bacillus thuringiensis*. At 4 h, the transcriptional activity of *Bacillus* CAZymes was associated with carbohydrate deacetylation and saccharification of cellulose and complex polysaccharides by LPMO (AA10). Cellulose is more recalcitrant than hemicellulose and showed lower degradation rates. LPMO activity may facilitate the action of hydrolytic enzymes, increasing the efficiency of cellulose degradation (37, 38). At 24 h, the expression of all CAZymes was downregulated. In this time, the concentration of soluble sugars could be incrementing the competition between members, benefiting microorganisms that do not effectively contribute to lignocellulose degradation (39).

In bacteria, the association of CBMs with different catalytic domains increments the functional diversity of CAZymes (40). The major proportion of CAZymes expressed by PM-06 exhibited modular architectures, which included one or more catalytic domains and CBMs. CBMs improve the catalytic activity of CAZymes by increasing enzyme

proximity to substrates, especially to insoluble ones (19, 40). In PM-06, the number of modular CAZymes and the architecture complexity increased at 120 h. CBMs could be enhancing the activity and increasing the penetration of enzymes into the NMP's more recalcitrant structures. Moreover, CBMs are also involved in the modulation of enzyme stability, function, and specificity (41). PM-06 expressed GH43, GH26, GH10, and GH5 at 0 and 120 h; however, these catalytic domains were associated with different CBMs, generating enzymes toward different substrates. These findings suggest that PM-06 is overcoming recalcitrance with a diversity of modular CAZymes operating in the proximity of insoluble substrates.

Microorganisms in PM-06 execute different but complementary metabolic tasks. *Paenibacillus* expressed modular enzymes with activity toward insoluble substrates; *Bacillus* expressed an LPMO with a CBM5 active on complex polysaccharides, such as crystalline cellulose and xylan (17), and CE that allow the access of enzymes into hemicellulose. *Microbacterium*, *Leifsonia*, and *Nocardia* expressed genes related to the hydrolysis of oligosaccharides. Moreover, the synergy between degraders and nondegraders is probably required to maintain microbial viability. Although the main focus of this study is lignocellulose degradation, the transcriptional activity of other abundant microorganisms in PM-06, such as *Aneurinibacillus*, could be related to the expression of genes with still unknown beneficial functions for the community. A metaproteomic study is needed to obtain more information about the proteins and enzymes produced by the consortium during degradation of NMP.

## MATERIALS AND METHODS

**PM-06 culture and sampling.** The consortium PM-06 was obtained by the dilution-to-stimulation approach (42) from the endogenous microbiota of the NMP. The initial culture (seed) was incubated for 7 days at 37°C with shaking at 125 rpm in flasks containing 25 mL of liquid medium (40 g/L NMP and 5 g/L yeast extract). At the end of the cultivation cycle (day 7), the consortium was inoculated in fresh medium under the same conditions. The metatranscriptomic analysis was carried out in the batch corresponding to the 10th transfer after evaluating its degrading stability and viability (data not shown). To obtain the genetic material, 5 mL of cell suspensions containing between 0.1 and 10 mg/mL cellular protein was taken directly from the fermentation broth and preserved in RNAprotect cell reagent (Qiagen) at −20°C before shipping. The samples were taken in triplicate (biological replicates) at the time of inoculation (0 h) and at 4, 24, and 120 h after inoculation.

**NMP degradation kinetics.** Degradation kinetics of NMP were performed as previously described (16). Differential degradation rates were calculated using the formula

$$\Delta W / \Delta t$$

where $W$ is the weight of solids consumed in the lapse of time defined by $t_2 - t_1$ ($\Delta t$).

**RNA isolation and sequencing.** Preserved PM-06 cell suspensions were processed for total RNA isolation and DNase treatment. Reverse transcription optimized for low-input RNA using SMARTer stranded total RNA-seq kit v2 (TaKaRa Bio USA, Inc.) was performed using treated RNA from all samples. Adapters were ligated for sequencing on the Illumina HiSeq2500 platform at a depth of 50 million paired-end reads (25 million per side). All procedures were carried out by Admera Health (NJ, USA).

Once the raw reads were obtained, adapters used in the sequencing were removed as well as sequences belonging to *Zea mays* (host organism). Filtered reads were processed with the FastQC tool, and all reads with a Phred score of <25 were discarded. Additionally, low-complexity sequences were identified and removed with Prinseq (43) to ensure the initial quality before proceeding with the assembly.

**Metatranscriptomic data processing and annotation.** Cross-assembly of post-quality control (post-QC) reads was performed with metaSPAdes (44) with default settings. High-quality reads were *de novo* assembled in 14,257 contigs. Open reading frame (ORF) calling was performed using Prodigal 2.60 with the meta procedure (45). Protein sequences were locally aligned by DIAMOND (46) against the NCBI nr database for taxonomic and functional annotation.

Metabolic annotation of predicted proteins was conducted using the GhostKOALA web platform (47). For CAZyme annotation, the protein sequences were processed with the automated CAZy annotation database (dbCAN) (48) in the dbCAN2 meta server (https://bcb.unl.edu/dbCAN2) using all search tools provided (HMMER, DIAMOND, and Hotpep).

**Genome recruitment.** Genome coverage or PM-06 transcripts in each stage of the NMP degradation process were estimated by a recruitment analysis. For the above, 12 publicly available complete microbial genomes (Table 2) were selected according to the most abundant species obtained from the taxonomic classification of protein-coding sequences. Metatranscriptomic reads in each sample were mapped against the genomes using the BLAT algorithm (49). Genome coverage of recruited reads was expressed in terms of the number of reads recruited per kilobase of genome per gigabase of

**TABLE 2** Complete bacterial genomes selected to be recruited by metatranscriptomic reads

| Organism | Genome size (bp) | Accession no.[a] |
|---|---|---|
| *Aneurinibacillus migulanus* DSM 2895 | 6,348,994 | LGUG01000000 |
| *Bacillus coagulans* S-lac | 3,694,837 | CP011939.1 |
| *Bacillus thuringiensis* YBT-1518 | 6,002,284 | CP005935.1 |
| *Leifsonia xyli* SE134 | 3,596,761 | CP014761.1 |
| *Microbacterium* sp. No. 7 | 4,599,046 | CP012697.1 |
| *Microbacterium* sp. LCT-H2 | 3,356,231 | MODW01000000 |
| *Nocardia nova* ATCC 33727 | 7,553,834 | PYHS01000000 |
| *Nocardia nova* SH22a | 8,348,532 | NZ_CP006850 |
| *Paenibacillus kribbensis* AM49 | 5,778,702 | CP020028.1 |
| *Paenibacillus macerans* 8224 | 7,331,450 | JMQA01000000 |
| *Paenibacillus macerans* FSL R5-0527 | 8,378,081 | MRTR01000000 |
| *Paenibacillus macerans* NCTC6355 | 7,390,297 | UGSI01000000 |

[a]All the genomes used are publicly available in the NCBI databases (https://www.ncbi.nlm.nih.gov).

metatranscriptome (RPKG) by counting the matching transcripts (hits) with a length of ≥50 bp, an identity of ≥95%, and an E value of ≤1E−5. Recruitment plots were generated to visualize genome coverage of strains (i.e., *Paenibacillus macerans* 8224 [Fig. S1 in the supplemental material]).

**Transcript abundance estimation and differential expression analysis.** All reads were mapped to the reference metatranscriptome and quantified using kallisto with default parameters (50). Raw counts were used as input for edgeR to perform pairwise statistical tests for differential expression between samples (51, 52). Transcripts that presented at least a $\log_2$ fold change of 2 at a false-discovery rate (FDR)-corrected *P* value cutoff of 0.001 in any of the pairwise statistical tests were considered differentially expressed. To visualize the contribution of microbial members to the degradation process, transcripts associated with CAZyme families from highly abundant microbial genera along with families classified as GH, AA, and PL were retrieved from the set of differentially expressed transcripts. Families were manually annotated according to the substrate in which they act.

To understand the role of specific CAZymes associated with lignocellulose degradation, families reported as relevant for this process (13, 14, 53–57) were retrieved from the previously filtered set. Transcripts were grouped according to their families, summing their corresponding expression values and obtaining the total log CPM for each CAZyme family. Finally, a *Z*-score was computed for each family across samples, followed by hierarchical clustering and visualization using the ComplexHeatmap library from Bioconductor (58). A *Z*-score of ≥0.4 was used as a threshold value for upregulation within the set of CAZyme transcripts analyzed.

**Data availability.** The reference metatranscriptome and raw reads for each sample were deposited in the Sequence Read Archive (SRA) database under BioProject accession number PRJNA735556.

## SUPPLEMENTAL MATERIAL

Supplemental material is available online only.
**SUPPLEMENTAL FILE 1**, XLSX file, 0.1 MB.
**SUPPLEMENTAL FILE 2**, PDF file, 0.7 MB.

## ACKNOWLEDGMENTS

This research was supported by CONACyT project CB 242952. J.G.S.-G. received a CONACyT doctoral fellowship. We thank Harinera de Yucatán and S.A. de C.V. for providing the nixtamalized maize pericarp. In addition, J.G.S.-G. thanks Francisco Rodríquez Valera for allowing a stay in the Evolutionary Genomics Group lab.

J.G.S.-G. developed the methodology, performed the formal analysis, and wrote, reviewed, and edited the paper. C.A.F.-E. developed the methodology and performed the formal analysis. P.J.C.-Y. performed the formal analysis, methodology, and data curation. F.H.C. contributed to data curation. R.A.R.-H. conceptualized the project and developed the methodology. M.N.S.-G. acquired funding, developed the methodology, conceptualized and supervised the project, performed the formal analysis, and reviewed and edited the paper.

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
