## [Reviewer comments · Microbiology Spectrum]

Microbiology Spectrum

Metatranscriptome profiling of a specialized microbial consortium during the degradation of nixtamalized maize pericarp

José Germán Serrano-Gamboa, Carlos Abraham Fócil-Espinosa, Pedro J. Cabello-Yeves, Felipe Coutinho, Rafael Antonio Rojas-Herrera, and Mónica Noel Sánchez-González

Corresponding Author(s): Mónica Noel Sánchez-González, Universidad Autónoma de Yucatán

Review Timeline:

Submission Date:

November 22, 2021

Accepted:

December 3, 2021

Editor: Jeffrey Gralnick

Reviewer(s): The reviewers have opted to remain anonymous.

Transaction Report:

DOI: <https://doi.org/10.1128/spectrum.02318-21>

December 3, 2021

Dr. Mónica Noel Sánchez-González
Universidad Autónoma de Yucatán
Facultad de Ingeniería Química
Periférico Norte Kilómetro 33.5
Tablaje catastral 13615 Chuburná de Hidalgo Inn
Mérida, Yucatán 97203
Mexico

Re: Spectrum02318-21 (Metatranscriptome profiling of a specialized microbial consortium during the degradation of nixtamalized maize pericarp)

Dear Dr. Mónica Noel Sánchez-González:

Based on your responses and revisions to the prior round of review, your manuscript has been accepted, and I am forwarding it to the ASM Journals Department for publication. You will be notified when your proofs are ready to be viewed.

Sincerely,

Jeffrey Gralnick
Editor, Microbiology Spectrum

Journals Department
Supplemental file 1: Accept
Data Set S1: Accept